# Don't Forget the Nonlinearity: Unlocking Activation Functions in Efficient Fine-Tuning

## Abstract

Existing parameter-efficient fine-tuning (PEFT) methods primarily adapt weight matrices while keeping activation functions fixed. We introduce **NoRA**, the first PEFT framework that directly adapts nonlinear activation functions in pretrained transformer-based models. NoRA replaces fixed activations with learnable rational functions and applies structured low-rank updates to numerator and denominator coefficients, with a group-wise design that localizes adaptation and improves stability at minimal cost. On vision transformers trained on CIFAR-10 and CIFAR-100, NoRA matches or exceeds full fine-tuning while updating only 0.4% of parameters (0.02M), achieving accuracy gains of +0.17% and +0.27%. When combined with LoRA (**NoRA++**), it outperforms LoRA and DoRA under matched training budgets by adding fewer trainable parameters. On LLaMA3-8B instruction tuning, NoRA++ consistently improves generation quality, yielding average MMLU gains of +0.3%–0.8%, including +1.6% on STEM (Alpaca) and +1.3% on OpenOrca. We further show that NoRA constrains adaptation to a low-dimensional functional subspace, implicitly regularizing update magnitude and direction. These results establish activation-space tuning as a complementary and highly parameter-efficient alternative to weight-based PEFT, positioning activation functions as first-class objects for model adaptation.

## 1 Introduction

Recent advances in deep learning have demonstrated the remarkable power of large-scale pretrained models across domains such as vision, language, and multimodal learning Abnar et al. (2021). However, deploying these models in downstream tasks often requires task-specific adaptation, posing significant challenges in terms of computational efficiency and parameter overhead Jiang et al. (2024); Lyu & Yin (2024). Full fine-tuning of all model parameters is not only costly but also prone to overfitting and catastrophic forgetting, especially when labeled data is limited or hardware resources are constrained Krizhevsky et al. (2009).

To address these issues, parameter-efficient fine-tuning (PEFT) Houlsby et al. (2019) techniques have emerged as a promising solution. Among them, Low-Rank Adaptation (LoRA) Hu et al. (2021) has gained significant attention by introducing trainable low-rank perturbations to frozen weight matrices, achieving strong performance with only a small fraction of trainable parameters. However, while these methods are effective for updating weight matrices, they largely overlook the potential of adapting non-linear components, such as activation functions. Existing PEFT approaches typically treat activation functions as fixed, immutable components, despite their crucial role in capturing task-specific inductive biases (e.g., smoothness, stability) Shi et al. (2024). This neglect of the adaptability of activation functions marks a critical gap in current PEFT strategies. Activations play a vital role in transforming input data at each layer of a neural network Sharma et al. (2017), and their adaptation is key to fine-tuning the model's performance for specific tasks. This shift from focusing solely on weights to also considering the adaptation of activations represents a fundamental rethinking of the PEFT paradigm, particularly in models like KANs Liu et al. (2025), where the activation functions themselves are learnable and dynamic.

In this work, we investigate fine-tuning strategies that target activation functions, using learnable rational functions as a flexible and expressive alternative to fixed nonlinearities. Unlike traditional architectures that rely on fixed nonlinearities, which remain static during training, our approach

leverages the fact that many widely-used activation functions such as ReLU Glorot et al. (2011), GELU Hendrycks & Gimpel (2016), and Swish Ramachandran et al. (2017), can be closely approximated or even exactly represented using rational functions. Models equipped with learnable rational activations replace these fixed nonlinearities with parameterized rational functions, allowing the nonlinear transformations to be expressed as:

$$\phi(x) = \frac{P(x)}{Q(x)} = \frac{\sum_{i=0}^{m} a_i x^i}{\sum_{j=0}^{n} b_j x^j},$$  (1)

where $\{a_i\}$ and $\{b_j\}$ are learnable coefficients. This insight implies that our method is theoretically applicable to any network by replacing fixed activations with their rational counterparts. However, adapting rational activations presents unique challenges: small perturbations in the denominator $Q(x)$ can lead to large functional changes or instability.

To overcome this challenge, we propose the **Nonlinear Rational Adapter (NoRA)**, the first parameter-efficient fine-tuning framework explicitly designed for the activation function components in model. NoRA first replaces the fixed activation functions with learnable rational functions, then introduces low-rank perturbations to both the numerator $P(x)$ and denominator $Q(x)$ coefficients, allowing task-specific adaptation while preserving the algebraic structure of rational transformations. By constraining updates to a structured low-dimensional subspace Nie et al. (2020), NoRA ensures smoothness, stability, and bounded functional deviation during fine-tuning—properties critical for the safe adaptation of rational activations.

**Our contributions are summarized as follows:**

- **A new paradigm for activation-centric PEFT:** We introduce NoRA, the first fine-tuning framework that directly targets the adaptation of activation functions. This shifts the focus of PEFT from weight matrices to the nonlinear components of neural networks.
- **Structured low-rank adaptation of rational functions:** NoRA perturbs both numerator and denominator coefficients in a theoretically grounded manner, preserving functional stability while enabling flexible task-specific adaptation.
- **Practical compatibility with rational activations:** NoRA complements existing rational function activations by providing a parameter-efficient adaptation mechanism. It operates without architectural changes, making it readily applicable across models that use rational approximations of standard nonlinearities.

By shifting the focus of PEFT from weight adaptation to activation-level adaptation, our work opens new directions for enhancing expressiveness and adaptability in modern neural architectures.

## 2 RELATED WORK

### 2.1 LOW RANK ADAPTATION (LORA)

Low-Rank Adaptation (LoRA) Hu et al. (2021) is a technique designed to efficiently fine-tune large pre-trained language models by reducing the number of trainable parameters. Instead of updating the entire weight matrix $W_0$ during training, LoRA introduces two low-rank matrices, $A$ and $B$, such that:

$$W = W_0 + \Delta W = W_0 + BA$$  (2)

Here, $W_0$ represents the original weight matrix, $\Delta W$ denotes the weight update, $B \in \mathbb{R}^{d \times r}$, and $A \in \mathbb{R}^{r \times k}$, where $r \ll \min(d, k)$ is the rank of the decomposition. This approach leverages the observation that the updates to the weights during model adaptation often have a low intrinsic rank, allowing for a significant reduction in the number of trainable parameters without compromising model performance.

During the forward pass, the output is computed as:

$$y = Wx = (W_0 + BA)x = W_0 x + BAx$$  (3)

In this formulation, $W_0$ remains fixed, and only the matrices $A$ and $B$ are updated during training. This strategy not only reduces computational and memory requirements but also mitigates issues such as catastrophic forgetting by preserving the original model parameters.

## 2.2 Learnable Activation Functions

Activation functions are critical to the expressivity and inductive biases of neural networks. While standard architectures rely on fixed nonlinearities such as ReLU Glorot et al. (2011), GELU Hendrycks & Gimpel (2016), and Swish Ramachandran et al. (2017), recent work has explored learnable activation functions that adapt their shape during training. Early parametric forms include PReLU He et al. (2015a), which learns a slope parameter for negative activations, and APL Agostinelli et al. (2015), which models activations as a piecewise-linear combination of hinge functions. Later developments use spline-based and kernel-based approximations for higher flexibility.

A particularly powerful family of learnable activations is based on rational functions. Due to their universal approximation property Baker Jr & Gammel (1961), rational functions can represent a wide range of continuous functions more compactly than polynomials. A rational activation is typically expressed as:

$$\phi(x) = \frac{P(x)}{Q(x)} = \frac{\sum_{i=0}^{m} a_i x^i}{\sum_{j=0}^{n} b_j x^j}, \tag{4}$$

where $\{a_i\}$ and $\{b_j\}$ are learnable coefficients of the numerator and denominator, respectively. This formulation allows activation functions to dynamically adjust their nonlinearity during training.

## 3 Nonlinear Rational Adapter (NoRA)

In this work, we propose the **Nonlinear Rational Adapter (NoRA)**, a novel parameter-efficient fine-tuning method that adapts pretrained models by modifying their nonlinear activation functions with learnable rational functions.

Traditional activation functions, such as ReLU Glorot et al. (2011) and GELU Hendrycks & Gimpel (2016), can all be approximately expressed as rational functions Telgarsky (2017) of the form:

$$\phi(x) = \frac{P(x)}{Q(x)} \tag{5}$$

where $P(x)$ and $Q(x)$ are polynomials of the form $P(x) = \sum_{i=0}^{m} a_i x^i$ and $Q(x) = \sum_{j=0}^{n} b_j x^j$, with learnable coefficients $\{a_i\}$ and $\{b_j\}$. Then this formulation also can be standardized to avoid division by zero as:

$$\phi(x) = \frac{a_0 + a_1 x + a_2 x^2 + \cdots + a_m x^m}{1 + |b_0 + b_1 x + b_2 x^2 + \cdots + b_n x^n|} \tag{6}$$

This formulation allows any fixed activation function to be represented as a rational function with learnable coefficients, providing more flexibility and expressiveness in modeling complex data transformations. While using a single shared rational activation function for all neurons limits the model's expressiveness, assigning a unique activation to each neuron is prohibitively expensive. Following Group-KAN Yang & Wang (2025), we partition the hidden (channel) dimension of each layer into $g$ disjoint groups, fixed across tokens and batches. All neurons in the same group share one learnable rational activation function. This static grouping preserves flexibility while keeping the overhead linear in $g$ (i.e., $g$ activations per layer) rather than per neuron.

Building on this idea, NoRA first replaces the fixed activation function with a group learnable rational function, and then injects structured low-rank perturbations Benaych-Georges & Nadakuditi (2011) into the coefficient matrices of these rational functions. Specifically, the perturbations are applied to both the numerator $P$ and the denominator $Q$ in a grouped fashion, where the coefficients are divided into $g$ groups. Let $\phi(X)$ denote the original rational function, and $\phi'(X)$ its perturbed counterpart. The resulting updated rational function is given by:

$$\phi'(X) = \frac{(P_g + \mathcal{L}_g(\Delta P))(X)}{(Q_g + \mathcal{L}_g(\Delta Q))(X)} \tag{7}$$

Here, $P_g$ and $Q_g$ represent the original polynomial numerator and denominator of the rational activation function, respectively. The perturbation terms $\Delta P$ and $\Delta Q$ are approximated using a group-wise low-rank adaptation function $\mathcal{L}_g(\cdot)$, which applies independent low-rank perturbations within each group $g = 1, \ldots, G$.

More concretely, the entire set of neurons is partitioned into $G$ disjoint groups, each containing $n/G$ neurons. For each group $g$, the adapted rational activation function is defined as

$$\phi_g'(X) = \frac{(P_g + A_g^P B_g^P)(X)}{(Q_g + A_g^Q B_g^Q)(X)} \tag{8}$$

where the perturbations $\Delta P_g = A_g^P B_g^P$ and $\Delta Q_g = A_g^Q B_g^Q$ are expressed as low-rank matrix products with

$$A_g^{(\cdot)} \in \mathbb{R}^{d \times r}, \quad B_g^{(\cdot)} \in \mathbb{R}^{r \times 1},$$

where $(\cdot) \in \{P, Q\}$, $d$ is the degree of the polynomial, and $r$ is the rank of the approximation.

All neurons within the same group $g$ share the adapted activation function $\phi_g'$, enabling parameter-efficient and localized functional adaptation. Increasing the number of groups $G$ enhances the granularity of adaptation, allowing more flexible modeling of complex activation patterns while keeping the parameter increase minimal due to the low-rank structure.

Furthermore, to ensure stable training and smooth fine-tuning from a pretrained baseline, all low-rank matrices are initialized similarly to LoRA Hu et al. (2021): $A_g$ is initialized with a small Gaussian noise (e.g., $\mathcal{N}(0, 0.02)$) and $B_g$ is initialized as zeros. This initialization guarantees that the adapted rational activation functions $\phi_g'(X)$ are equivalent to the original functions $P_g(X)/Q_g(X)$ at the start of training. The reason why tuning activation matters and why we choose rational function are shown in Appendix A, B

An overview of the NoRA framework is illustrated in Figure 1.

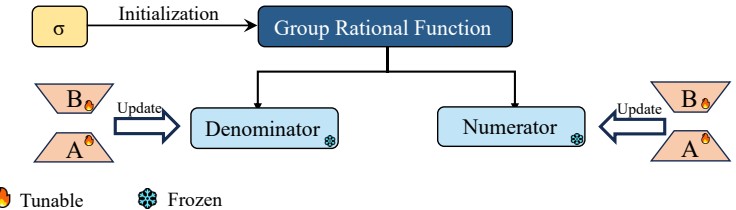

Figure 1: Overview of the NoRA framework. NoRA replaces fixed activation functions with group rational functions and introduces structured low-rank perturbations to both the numerator and denominator coefficients.

## 4 EXPERIMENT

In this section, we evaluate the performance of NoRA across both image classification and language model tasks. Specifically, we apply NoRA to the ViT-Tiny model for CIFAR-10 and CIFAR-100 classification, and to the LLaMA3-8B model for instruction tuning.

### 4.1 COMPARISON WITH PARAMETER-EFFICIENT FINE-TUNING METHODS IN IMAGE CLASSIFICATION

#### 4.1.1 EXPERIMENT SETUP.

We conduct experiments using the ViT-Tiny model pretrained on ImageNet-1K Deng et al. (2009) For adaptation, we explore two PEFT configurations: (1) **NoRA**, where we only replace the GELU in FFN with the rational activation function and use group-wise low-rank perturbations while keeping all other weights frozen; and (2) **NoRA++**, a hybrid variant that combines NoRA with standard LoRA applied to the MLP layers in attention layers. In NoRA++, both the activation functions

Table 1: Comparison with parameter-efficient fine-tuning methods on CIFAR-10 and CIFAR-100.

| Method | Trainable Params (M) | CIFAR-10 Acc. (%) | CIFAR-100 Acc. (%) |
|---|---|---|---|
| Full tuning | 5.54 (100%) | 90.71 | 77.19 |
| VPT | 0.39 (7.0%) | 89.62 (−1.09) | 75.43 (−1.76) |
| Adapter | 0.48 (8.7%) | 89.93 (−0.78) | 75.88 (−1.31) |
| LoRA | 0.33 (6.0%) | 91.05 (+0.34) | 77.68 (+0.49) |
| QLoRA | 0.33 (6.0%) | 90.45 (−0.26) | 77.01 (−0.18) |
| DoRA | 0.34 (6.1%) | 91.13 (+0.42) | 77.71 (+0.52) |
| **NoRA (ours)** | 0.02 (0.4%) | 90.88 (+0.17) | 77.46 (+0.27) |
| **NoRA++ (ours)** | 0.35 (6.2%) | **91.24** (+0.53) | **77.76** (+0.57) |

and select linear weights are jointly adapted, offering a more expressive yet still parameter-efficient fine-tuning scheme. For both NoRA and NoRA++, the low-rank perturbation rank is set to $r = 2$. In both settings, the classification head is also trained. Evaluation is performed on CIFAR-10 and CIFAR-100 Krizhevsky et al. (2009), two widely used image classification benchmarks. CIFAR-10 includes 10 object classes, while CIFAR-100 contains 100 fine-grained classes grouped into 20 superclasses. Each dataset provides 6000 or 600 samples per class, with image resolution $32 \times 32$. We resize images to $224 \times 224$ and use a patch size of 16 Dosovitskiy et al. (2020) during training. The specific hyperparameter settings can be referred to in Appendix C.

### 4.1.2 BASELINE METHODS.

To provide a comprehensive evaluation, we compare NoRA with several representative parameter-efficient fine-tuning (PEFT) methods. These include **Full Fine-Tuning**, which updates all model parameters and serves as an upper-bound reference; **VPT** Jia et al. (2022), which prepends learnable visual prompt tokens to the input sequence while keeping the backbone frozen; **Adapter** Chen et al. (2022), which inserts lightweight bottleneck modules between transformer blocks and updates only these modules during training; **LoRA** Hu et al. (2021), which introduces trainable low-rank matrices into attention layers while freezing the original weights; **QLoRA** Dettmers et al. (2023), which extends LoRA to 4-bit quantized models for memory-efficient adaptation; and **DoRA** Liu et al. (2024), which decomposes pre-trained weights into magnitude and direction components to better approximate the behavior of full fine-tuning. All methods are implemented on the same ViT-Tiny backbone for fair comparison, with the classification head remaining trainable. Detailed hyperparameter settings are provided in Appendix C.1.

### 4.1.3 RESULT ANALYSIS

As shown in Table 1, NoRA, while tuning only 0.02M parameters (0.4%), achieves 90.88% accuracy on CIFAR-10 and 77.46% on CIFAR-100, outperforming full fine-tuning by +0.17% and +0.27%, respectively. This highlights the surprising effectiveness of adaptively tuning activation functions alone, without modifying any backbone weights. In contrast, other PEFT baselines such as LoRA and DoRA also slightly surpass full fine-tuning but require over 6% of the model parameters to be updated—more than $15\times$ as many as NoRA. Meanwhile, Adapter, QLoRA, and VPT lag behind in both accuracy and efficiency, underscoring the importance of adaptation position and mechanism. To explore composability, we further introduce a hybrid variant, **NoRA++**, which applies NoRA to the activation functions and LoRA to the attention and MLP layers. This integration yields the best accuracy on both datasets—91.24% on CIFAR-10 and 77.76% on CIFAR-100—while still using fewer trainable parameters than full fine-tuning. These results confirm that NoRA offers an excellent trade-off between accuracy and efficiency, and its compatibility with weight-based methods like LoRA enables scalable and flexible adaptation strategies.

### 4.2 SCALABILITY WITH GROUP EXPANSION.

### 4.2.1 EXPERIMENT SETUP.

To further investigate the capacity and scalability of NoRA, we systematically increase the number of groups $g$ by powers of two, setting $g = 8, 16, 32, 64$, while keeping the rank fixed at $r = 3$ and all

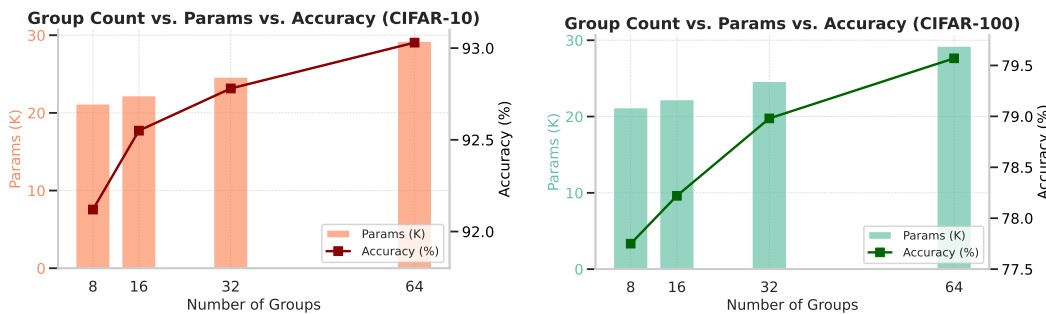

Figure 2: The relationship curve between the number of groups and the number of parameters(without classification head) and the classification accuracy of CIFAR-10/100 during fine-tuning.

other training settings unchanged. This allows us to study how finer group partitioning influences the expressiveness and adaptability of the rational activation functions. To initialize the parameters for these finer group divisions efficiently, we adopt a simple replication strategy, wherein the original group-wise rational coefficients are duplicated along the channel dimension to match the increased number of groups. This approach enables higher-resolution activation adaptation without modifying the model architecture, thus providing a practical and scalable way to enhance NoRA's functional flexibility.

### 4.2.2 RESULT ANALYSIS.

This straightforward strategy yields consistent performance improvements, as illustrated in Figure 2. As the number of groups $g$ increases from 8 to 64, NoRA achieves progressively higher accuracy on both CIFAR-10 (from approximately 92.2% to 93.1%) and CIFAR-100 (from approximately 77.6% to 79.6%). Meanwhile, the number of trainable parameters increases only moderately (from approximately 13K to 28K), reflecting the controlled growth enabled by the low-rank group-wise design. These results empirically confirm that increasing group resolution facilitates more localized and specialized nonlinear modeling, allowing NoRA to better capture task-specific activation dynamics. The dual-axis plots in Figure 2 clearly illustrate this trade-off: performance scales almost linearly with group count, while parameter cost grows sub-linearly, underscoring NoRA's efficiency. Notably, these gains are achieved without tuning additional hyperparameters or introducing significant training overhead, further highlighting the practicality and scalability of the proposed method. This behavior supports the intuition that composing structured local approximations can effectively approximate global nonlinear functions within the activation space.

### 4.3 INSTRUCT-TUNING IN LARGE LANGUAGE MODEL

### 4.3.1 EXPERIMENT SETUP.

To assess the compatibility and enhanced effectiveness of our method in joint instruction-tuning settings, we introduce NoRA++, a hybrid adaptation framework that combines the proposed activation-centric fine-tuning with conventional weight-space tuning. Specifically, we integrate NoRA with LoRA on the LLaMA3-8B model by replacing the fixed activation functions in the MLP blocks of each Transformer layer with group-wise rational functions, and then applying structured low-rank adaptation via NoRA. This modification enables activation-space tuning while maintaining the parameter efficiency of LoRA's weight-space updates, resulting in a complementary and synergistic tuning mechanism. We evaluate NoRA++ on a suite of five diverse instruction datasets—*Alpaca*, *MathInstruct*, *OpenOrca*, *ShareGPT-Hyper*, and *UltraChat*—encompassing tasks from open-ended dialogue and reasoning to summarization. Under identical training budgets, NoRA++ consistently surpasses standard LoRA across all datasets, achieving significant improvements in output quality and generalization. To quantitatively assess its reasoning ability, we report results on the MMLU Hendrycks et al. (2020) benchmark under a 5-shot evaluation setup. NoRA++ yields consistent gains of +0.3% to +0.8% in average accuracy over LoRA, demonstrating that activation-centric

adaptation provides meaningful benefits even when layered atop established PEFT methods. Full implementation and hyperparameter details are provided in Appendix C .

### 4.3.2 RESULT ANALYSIS.

As shown in Table 2, the combination of NoRA and LoRA yields consistent and measurable performance gains across most instruction-tuning datasets and MMLU categories. For example, on Alpaca and MathInstruct, NoRA++ improves the average accuracy by +0.8% and +0.5%, respectively, with notable gains such as +1.6% in STEM (Alpaca) and +2.3% in STEM (MathInstruct). Even on OpenOrca, which already benefits from strong LoRA tuning, the addition of NoRA leads to further improvements in key areas like STEM (+1.3%) and Social Sciences (+1.2%), resulting in a net average increase. While a few categories exhibit minor regressions—for instance, –0.7% in Humanities on OpenOrca and –0.4% in the "Other" category of ShareGPT-Hyper—the overall trend remains positive. These results suggest that NoRA effectively introduces complementary local nonlinearity at the activation level, enhancing model expressiveness without disrupting the core low-rank structure imposed by LoRA. This synergy between activation-level and weight-level tuning highlights NoRA's general applicability and its potential as a plug-and-play enhancement module for a wide range of parameter-efficient fine-tuning (PEFT) methods in large-scale language models.

Table 2: MMLU-test accuracy (%) after instruction tuning LLaMA3-8B on five datasets. Each cell shows LoRA result followed by NoRA+LoRA result and delta. Green indicates improvement and red indicating decline.

| Tuning Dataset | STEM | Humanities | Social Sciences | Other | Average |
|---|---|---|---|---|---|
| Alpaca | $60.2 \rightarrow$ **61.8** (+1.6) | $65.5 \rightarrow$ **65.0** (–0.5) | $63.3 \rightarrow$ **64.5** (+1.2) | $62.0 \rightarrow$ **63.0** (+1.0) | $62.8 \rightarrow$ **63.6** (+0.8) |
| MathInstruct | $53.7 \rightarrow$ **56.0** (+2.3) | $75.5 \rightarrow$ **74.9** (–0.6) | $58.3 \rightarrow$ **59.2** (+0.9) | $70.7 \rightarrow$ **70.4** (–0.3) | $63.9 \rightarrow$ **64.4** (+0.5) |
| OpenOrca | $54.2 \rightarrow$ **55.5** (+1.3) | $74.6 \rightarrow$ **73.9** (–0.7) | $58.1 \rightarrow$ **59.3** (+1.2) | $71.0 \rightarrow$ **70.4** (–0.6) | $63.9 \rightarrow$ **64.2** (+0.3) |
| ShareGPT-Hyper | $58.9 \rightarrow$ **59.2** (+0.3) | $66.3 \rightarrow$ **67.0** (+0.7) | $60.7 \rightarrow$ **61.0** (+0.3) | $59.4 \rightarrow$ **59.0** (–0.4) | $61.3 \rightarrow$ **61.6** (+0.3) |
| UltraChat | $57.2 \rightarrow$ **57.9** (+0.7) | $64.8 \rightarrow$ **65.3** (+0.5) | $60.2 \rightarrow$ **59.8** (–0.4) | $59.7 \rightarrow$ **60.6** (+0.9) | $60.5 \rightarrow$ **60.9** (+0.4) |

## 4.4 ABLATION STUDY AND ANALYSIS

In this section we present four experiments to assess rational activations, low-rank perturbations, selective coefficient tuning, and overall efficiency.

### 4.4.1 COMPARISON WITH OTHER LEARNABLE ACTIVATIONS.

To validate the necessity of employing rational functions as activation mechanisms, we perform an ablation study comparing NoRA with several common learnable activation functions. Specifically, we replace the GELU Hendrycks & Gimpel (2016) activations in the MLP layers of the pretrained ViT-Tiny model with alternative nonlinearities, including **PReLU** He et al. (2015b), **AReLU** Chen et al. (2020), and **ELU** Clevert et al. (2016). We then fine-tune only the activation function parameters and the classification head on CIFAR-100, while keeping

Table 3: Ablation on learnable activation functions.

| Name | Accuracy (%) |
|---|---|
| PReLU | 53.21 |
| AReLU | 54.17 |
| ELU | 52.84 |
| **NoRA (Ours)** | **77.46** |

all other model weights frozen. As reported in Table 3, these general-purpose activation functions yield only marginal improvements, with top-1 accuracy consistently lagging behind our proposed rational activation-based approach. This suggests that simple substitution of activation functions fails to provide sufficient task-specific adaptability or structural compatibility within the frozen transformer architecture. In contrast, our method enables structured, localized, and task-adaptive modulation of the activation landscape through low-rank perturbations of rational functions, yielding superior representational refinement under stringent parameter constraints.

### 4.4.2 DIFFERENT TUNING METHODS FOR LEARNABLE RATIONAL ACTIVATIONS.

To evaluate the effectiveness of NoRA, we compare it with several fine-tuning strategies that all keep the backbone frozen and only update the classification head along with the learnable rational

functions on CIFAR-100. **Rational fine-tuning** directly updates the coefficients of the rational activation functions. **Zero-initialized fine-tuning** introduces a learnable matrix initialized to zero and added to the rational coefficients. **GELU-initialized fine-tuning** initializes the rational functions to approximate GELU before training. **Const-tuning (a0,b0 only)** restricts adaptation to the constant terms of the numerator and denominator, i.e., updating only $a_0$ and $b_0$, which controls global offset/scale but leaves the nonlinear shape fixed. As shown in Table 4, all three baselines underperform, suggesting that coefficient-only tuning lacks sufficient expressiveness. In contrast, NoRA achieves the highest accuracy by introducing a learnable low-rank shift in the activation space, offering more flexible and effective adaptation.

### 4.4.3 IMPACT OF SELECTIVE PERTURBATION ON RATIONAL COEFFICIENTS.

To further investigate the importance of jointly adapting both components of the rational activation, we conduct an ablation study in which low-rank perturbations are selectively applied to either the numerator or the denominator coefficients, while keeping the other component fixed. As presented in Table 5, perturbing only one side leads to a noticeable performance degradation on both CIFAR-10 and CIFAR-100, compared to the setting where both components are jointly adapted. This result suggests that the effectiveness of NoRA stems not from modulating a single part of the activation function, but from the synergistic interaction between the numerator and denominator. Specifically, the numerator controls the functional shape of the activation, while the denominator governs numerical stability and saturation behavior. Their co-adaptation introduces a richer and more flexible activation landscape, which is critical for the improved generalization performance observed. These findings underscore the necessity of NoRA's co-perturbation design and highlight a fundamental architectural distinction from existing parameter-efficient fine-tuning methods.

Table 4: Ablation on different fine-tuning strategies.

| Method | Accuracy (%) |
|---|---|
| Rational tuning | 76.56 |
| Zero-init tuning | 76.44 |
| GELU-init tuning | 76.07 |
| Const-tuning ($a_0, b_0$ only) | 74.91 |
| **NoRA (ours)** | **77.46** |

| Numerator | Denominator | CIFAR-10 Acc. (%) | CIFAR-100 Acc. (%) |
|---|---|---|---|
| ✗ | ✓ | 87.40 | 74.16 |
| ✓ | ✗ | 86.92 | 73.58 |
| ✓ | ✓ | **90.88** | **77.46** |

Table 5: Effect of selectively injecting low-rank perturbations into numerator and denominator coefficients. ✓ indicates perturbed (trainable), ✗ indicates unperturbed (frozen).

### 4.4.4 RESOURCE EFFICIENCY ANALYSIS.

We assess the resource efficiency of NoRA and LoRA from three key perspectives: inference-time computational cost (FLOPs), latency, and the number of trainable parameters, all evaluated on the ViT-Tiny backbone. As shown in Table 6, NoRA achieves over $17\times$ reduction in trainable parameter count compared to LoRA (2.10K vs. 37.24K), highlighting its extreme parameter efficiency. Despite this, the inference latency remains comparable—5.69 ms/sample for NoRA vs. 5.30 ms/sample for LoRA—indicating that the additional expressiveness introduced by activation modulation does not impose substantial runtime overhead. The slight increase in FLOPs (1.07G vs. 0.91G) is expected, as NoRA introduces group-wise rational activation functions, which involve evaluating both polynomial numerators and denominators across multiple activation groups. This minor computational overhead is a direct result of NoRA's more flexible nonlinear modeling capability, and reflects the trade-off between functional expressiveness and cost. Importantly, the trainable parameter counts reported here exclude the classification head to ensure fair comparison. Overall, these results reinforce NoRA's strength as a lightweight and deployment-friendly PEFT method, with minimal runtime cost and strong potential for integration with other techniques like LoRA or adapters.

| Method | Params (K) | FLOPs (G) | Inference Time (ms/sample) |
|---|---|---|---|
| LoRA | 37.24 | 0.91 | 5.30 |
| **NoRA (ours)** | 2.10 | 1.07 | 5.69 |

Table 6: Comparison of resource efficiency between LoRA and NoRA on ViT-Tiny.

## 5 CONCLUSION AND FUTURE WORK

### 5.1 CONCLUSION

In this work, we proposed the **Nonlinear Rational Adapter (NoRA)**, a novel and general framework for parameter-efficient fine-tuning that shifts the focus of adaptation from the traditional weight-centric view to a new activation-centric perspective. By replacing fixed nonlinearities with task-adaptive rational functions, NoRA enables flexible and expressive modulation of pretrained models through compact, structured perturbations applied directly in the activation space. This is achieved via learnable low-rank parameterizations of both the numerator and denominator of rational functions, allowing for stable, efficient, and interpretable task adaptation while preserving the overall model architecture. The proposed group-wise formulation further enhances scalability by localizing adaptation across independently modulated activation units, thereby reducing parameter redundancy and improving representational flexibility. Extensive experiments on image classification (CIFAR-10/100) and instruction tuning of large language models (LLaMA3-8B) demonstrate that NoRA significantly outperforms full fine-tuning, despite updating only 0.4% of the total model parameters. Moreover, our extended variant, NoRA++, which integrates NoRA with LoRA for joint adaptation of activations and weights, achieves even stronger performance, consistently surpassing both DoRA and LoRA across vision and language tasks while maintaining superior parameter efficiency. Collectively, these results validate activation-centric adaptation as a powerful and underexplored dimension in the fine-tuning landscape, offering a complementary perspective to weight-based PEFT approaches and paving the way for more modular, robust, and efficient model customization strategies within large-scale pretraining paradigms.

### 5.2 FUTURE WORK

We plan to extend NoRA beyond classification and instruction tuning to more complex architectures, including generative diffusion models, graph neural networks, and vision and language models, where structured nonlinearity may be particularly beneficial. We will study functional classes beyond rationals (for example spline based, Fourier inspired, or attention conditioned activations) to clarify expressiveness and efficiency tradeoffs. We will develop adaptive group wise strategies that automatically select the number of groups $g$ and the subspace rank $r$ during training, which enables dynamic control of capacity and cost. We will also integrate NoRA with complementary PEFT methods such as prompt tuning, adapters, and LoRA variants across different network levels to achieve compositional and task aware adaptation. Finally, we aim to scale NoRA to trillion parameter language models and to evaluate it on long context, multi hop reasoning, and multi task benchmarks. Because NoRA operates in activation space and is modular with respect to weight centric updates, it can be inserted into existing pipelines and is suitable for deployment under resource or privacy constraints, including edge inference, online learning, and user personalization, where adapting lightweight nonlinear components enables efficient and stable continual learning on dynamic or streaming data.

## ETHICS STATEMENT

This work adheres to the ICLR Code of Ethics. Our study does not involve human subjects, sensitive data, or potentially harmful applications. All experiments were conducted using publicly available datasets, and there are no conflicts of interest or ethical concerns associated with this research.

## REPRODUCIBILITY STATEMENT

We provide detailed descriptions of datasets, experimental settings, and model hyperparameters in the main text and appendix. All algorithms and theoretical claims are fully documented, and code to reproduce the results will be made publicly available upon publication.

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

# A   WHY TUNING ACTIVATION MATTERS?

**Setup and notation.**   Consider a depth-$L$ network with frozen weight matrices $W_1, \ldots, W_L$ and elementwise, possibly group-shared, activations $\phi_\ell(\cdot; \theta_\ell)$:

$$h_0(x) = x, \quad z_\ell = W_\ell h_{\ell-1}(x), \quad h_\ell(x) = \phi_\ell(z_\ell; \theta_\ell), \quad F(x) = h_L(x). \tag{9}$$

Here $\theta_\ell \in \mathbb{R}^{p_\ell}$ are *activation parameters*. We analyze why adapting $\{\theta_\ell\}$ (while freezing $\{W_\ell\}$) is impactful for expressivity, stability, optimization, and generalization.

## A.1   FUNCTIONAL DIRECTIONS UNLOCKED BY ACTIVATION PARAMETERS

Let $\Delta\theta = \{\Delta\theta_\ell\}_{\ell=1}^L$ be a small update. By the chain rule,

$$F(x; \theta + \Delta\theta) - F(x; \theta) = \sum_{\ell=1}^L \underbrace{\frac{\partial F}{\partial h_\ell}(x)}_{\Pi_{j=\ell+1}^L J_j(x)} \cdot \underbrace{\frac{\partial h_\ell}{\partial \theta_\ell}(x)}_{D_\ell(x)\,\Delta\theta_\ell} + \mathcal{O}(\|\Delta\theta\|^2). \tag{10}$$

We denote

$$J_j(x) := \mathrm{Diag}\big(\phi_j'(z_j; \theta_j)\big)\, W_j, \tag{11}$$

and

$$D_\ell(x) \in \mathbb{R}^{d_\ell \times p_\ell}, \quad [D_\ell(x)]_{i,:} = \Big[\frac{\partial \phi_\ell(z_{\ell,i}; \theta_\ell)}{\partial \theta_\ell}\Big]^\top. \tag{12}$$

Thus the *first-order functional change* lies in the span

$$\Delta F(\cdot) \in \mathrm{span}\Big\{ \big(\Pi_{j=\ell+1}^L J_j(\cdot)\big) D_\ell(\cdot)\, e_k \ : \ 1 \le \ell \le L, \ 1 \le k \le p_\ell \Big\}. \tag{13}$$

Its intrinsic dimension is at most $\sum_\ell p_\ell$ (or $\sum_\ell G_\ell p_\ell$ if parameters are shared across $G_\ell$ groups). *Therefore, activation tuning provides a low-dimensional yet* function-space *set of directions unavailable if activations are fixed.*

## A.2   STABILITY AND LIPSCHITZ CONTROL

Denote the Lipschitz seminorm of $\phi_\ell(\cdot; \theta_\ell)$ as

$$\mathrm{Lip}(\phi_\ell; \theta_\ell) := \sup_z \big|\phi_\ell'(z; \theta_\ell)\big|. \tag{14}$$

For each block $h_\ell = \phi_\ell \circ W_\ell$,

$$\mathrm{Lip}(h_\ell; \theta_\ell) \le \mathrm{Lip}(\phi_\ell; \theta_\ell)\,\|W_\ell\|_2. \tag{15}$$

Hence the network Lipschitz constant satisfies

$$\mathrm{Lip}(F; \theta) \le \prod_{\ell=1}^L \mathrm{Lip}(\phi_\ell; \theta_\ell)\,\|W_\ell\|_2. \tag{16}$$

**Theorem A.1** (Network-level deviation under activation updates). *Let $F'$ denote the network after changing $\theta \mapsto \theta + \Delta\theta$. Then for any $x$,*

$$\|F'(x) - F(x)\|_2 \le \sum_{\ell=1}^L \Big(\prod_{j > \ell} \mathrm{Lip}(h_j; \theta_j)\Big) \|\Delta\phi_\ell(z_\ell)\|_2, \tag{17}$$

*where*

$$\Delta\phi_\ell(z) := \phi_\ell(z; \theta_\ell + \Delta\theta_\ell) - \phi_\ell(z; \theta_\ell). \tag{18}$$

*Moreover,*

$$\|\Delta\phi_\ell(z_\ell)\|_2 \le \sup_z \|D_\ell(z)\|_{2\to2}\,\|\Delta\theta_\ell\|_2\,\sqrt{d_\ell} + \mathcal{O}(\|\Delta\theta_\ell\|_2^2). \tag{19}$$

*Consequences.* (i) By directly controlling $\mathrm{Lip}(\phi_\ell; \theta_\ell)$ one can regularize the global Lipschitz constant, which enters standard generalization and robustness bounds. (ii) The deviation bound depends linearly (first order) on activation parameters through $D_\ell$, enabling fine-grained, stable modulation even with frozen weights.

### A.3 GRADIENT FLOW, JACOBIANS, AND EFFECTIVE RANK

For a single block $h = \phi(Wx; \theta)$, the input Jacobian is

$$J_x = \frac{\partial h}{\partial x} = \text{Diag}(\phi'(Wx; \theta)) \, W. \tag{20}$$

Activation parameters change $J_x$ via $\phi'$, thus altering (i) sensitivity to inputs, (ii) conditioning of the layer, and (iii) gradient flow to preceding layers. If $\phi$ is gated (e.g., with a slope/threshold parameter), let $\mathcal{A}(\theta) := \{i : \phi'(z_i; \theta) > 0\}$ be the active set. Then

$$Rank(J_x) \leq |\mathcal{A}(\theta)| \leq d. \tag{21}$$

Hence, tuning $\theta$ modulates the effective rank and spectrum of the linearized map $\Pi_\ell J_\ell$, improving gradient propagation in deep stacks.

### A.4 OPTIMIZATION GEOMETRY AND THE NTK PERSPECTIVE

Let $\vartheta := (\{W_\ell\}, \{\theta_\ell\})$ and write the neural tangent kernel (NTK) $K_\vartheta(x, x') = \langle \nabla_\vartheta F(x), \nabla_\vartheta F(x') \rangle$. Decompose

$$K_\vartheta(x, x') = K_{WW}(x, x') + K_{\theta\theta}(x, x') + 2K_{W\theta}(x, x'). \tag{22}$$

Using the first-order expansion,

$$\nabla_{\theta_\ell} F(x) = \left( \Pi_{j > \ell} J_j(x) \right) D_\ell(x), \tag{23}$$

so $K_{\theta\theta}$ spans activation-feature directions $\{D_\ell\}$ propagated to the output.

**Proposition A.1** (Complementarity at initialization). *Suppose pre-activations $z_\ell$ are centered and whitened within groups, and the initial $W_\ell$ are independent of $\theta_\ell$ (with zero-mean entries). Then*

$$\mathbb{E} \, K_{W\theta}(x, x') = 0, \qquad \mathbb{E} \, K_\vartheta(x, x') = \mathbb{E} \, K_{WW}(x, x') + \mathbb{E} \, K_{\theta\theta}(x, x'). \tag{24}$$

### A.5 CURVATURE CONTROL VIA $\phi'$ AND $\phi''$

Let $\mathcal{L}$ be a twice-differentiable loss and denote $J_x = \partial h / \partial x$. For block $h = \phi(Wx; \theta)$,

$$\frac{\partial^2 h}{\partial x^2} = \text{Diag}(\phi''(Wx; \theta)) \, [Wx] \, [Wx]^\top \odot I. \tag{25}$$

Hence the input Hessian of the loss obeys

$$\nabla_x^2 \mathcal{L} = J_x^\top \nabla_h^2 \mathcal{L} \, J_x + \sum_i \frac{\partial \mathcal{L}}{\partial h_i} \frac{\partial^2 h_i}{\partial x^2}. \tag{26}$$

Tuning $\theta$ therefore changes both the Gauss–Newton part (through $\phi'$ in $J_x$) and the residual curvature term (through $\phi''$), reshaping the local optimization landscape without touching $W$.

### A.6 DATA-ALIGNED GRADIENTS AND SATURATION AVOIDANCE

Let the gradient w.r.t. a preceding weight matrix $W_\ell$ be

$$\nabla_{W_\ell} \mathcal{L} = \left( \delta_\ell \odot \phi_\ell'(z_\ell; \theta_\ell) \right) h_{\ell-1}(x)^\top, \tag{27}$$

where $\delta_\ell$ is the backpropagated error. Then

$$\mathbb{E} \, \|\nabla_{W_\ell} \mathcal{L}\|_F^2 = \mathbb{E} \left[ \|\delta_\ell\|_2^2 \, \|h_{\ell-1}\|_2^2 \right] \cdot \mathbb{E} \, \overline{\phi_\ell'(z_\ell; \theta_\ell)^2}, \tag{28}$$

where the overline denotes the average across units. Choosing $\theta_\ell$ to *maximize derivative mass* under the moments of $z_\ell$ keeps most units responsive (not saturated), improving gradient signal-to-noise without changing $W_\ell$.

## A.7 GENERALIZATION VIA CAPACITY CONTROL

Define the hypothesis class $\mathcal{F}$ with frozen $\{W_\ell\}$ and activation parameters bounded by $\|\theta_\ell\| \leq \rho_\ell$ and local Lipschitz budgets $\mathrm{Lip}(\phi_\ell; \theta_\ell) \leq \lambda_\ell$. Using standard Lipschitz-based complexity bounds, for inputs $\|x\| \leq R$ and $N$ samples,

$$\mathfrak{R}_N(\mathcal{F}) \;\lesssim\; \frac{R}{\sqrt{N}} \cdot \left( \sum_{\ell=1}^{L} \Big( \prod_{j>\ell} \lambda_j \|W_j\|_2 \Big) \cdot c_\ell(\rho_\ell)\, \|W_\ell\|_2 \right), \tag{29}$$

where $c_\ell(\rho_\ell)$ is a polynomial-in-$\rho_\ell$ constant induced by the chosen parametrization of $\phi_\ell$.

**Takeaways.**

- **Functional expressivity at low cost.** Activation parameters open a low-dimensional *function-space* of directions (Eq. 13).
- **Stable, controllable modulation.** Activation Lipschitz and curvature ($\phi'$, $\phi''$) yield explicit network-level deviation and robustness control (Eqs. 16, 17, 26).
- **Better conditioning and gradient flow.** By changing gates/slopes, activation tuning controls Jacobian rank/spectrum and mitigates saturation (Eqs. 20, 21, 28).
- **Complementary optimization geometry.** Activation-parameter gradients contribute an NTK component largely orthogonal (in expectation) to weight-only directions (Eqs. 22, 23, 24).

These results justify *activation tuning* as a principled and effective axis for adapting pretrained models, independently of the particular parameterization chosen for $\phi_\ell$.

# B WHY RATIONAL FUNCTION?

Rational functions offer a compact and flexible parametrization that can *uniformly approximate* the activation functions used in modern networks—both smooth (e.g., $\tanh$, sigmoid, GELU/erf-based, SiLU/Swish) and non-smooth (e.g., ReLU, Leaky/ReLU)—on any bounded pre-activation domain. We record the key facts concisely.

**Definition.** A degree-$(m, n)$ rational function is

$$r_{m,n}(x) \;=\; \frac{P_m(x)}{Q_n(x)} \;=\; \frac{\sum_{i=0}^{m} a_i x^i}{\sum_{j=0}^{n} b_j x^j}, \tag{30}$$

and we assume the domain $K \subset \mathbb{R}$ is compact with a pole-free margin

$$\inf_{x \in K} |Q_n(x)| \;\geq\; \gamma \;>\; 0, \tag{31}$$

which is standard for stable approximation on $K$.

**Density on compact sets.** Because polynomials are dense in $C(K)$ (Stone–Weierstrass) and polynomials are a special case of rationals (take $Q_n \equiv 1$), rationals are also dense:

$$\forall f \in C(K),\ \forall \varepsilon > 0,\ \exists\, m, n,\ \exists\, r_{m,n} \text{ s.t. } \sup_{x \in K} |f(x) - r_{m,n}(x)| \;<\; \varepsilon. \tag{32}$$

Thus any continuous activation used in practice admits uniform rational approximations on bounded pre-activation ranges.

**Fast rates for smooth activations.** When $f$ is real-analytic on a neighborhood of $K$ (typical for sigmoid/$\tanh$/erf-like activations), best rational approximants achieve geometric convergence:

$$\exists\, C > 0,\ \rho > 1:\quad \inf_{\deg(r) \leq N} \sup_{x \in K} |f(x) - r(x)| \;\leq\; C\, \rho^{-N}, \tag{33}$$

where $N = m + n$ is total degree. This is substantially faster than the algebraic rates of many polynomial schemes.

**Near-root-exponential rates for kinks.** For non-smooth activations with a finite number of kinks (e.g., ReLU, $|x|$), rational approximation still excels:

$$\exists\, C, c > 0: \quad \inf_{\deg(r) \leq N}\, \sup_{x \in K} |f(x) - r(x)| \ \leq\ C \exp\big(-c\sqrt{N}\big), \qquad (34)$$

whereas the best polynomial error decays only algebraically in $N$. Hence even piecewise-linear activations can be approximated to high accuracy with modest rational degree.

**Practical consequences.**

- **Coverage.** Eqs. equation 32–equation 34 ensure a single rational family can approximate most activations used in deep learning on bounded pre-activation sets.
- **Efficiency.** The fast rates in equation 33–equation 34 imply that low degrees suffice in practice, keeping parameters and compute small.
- **Stability.** The pole-free margin equation 31 ensures bounded slopes/curvatures on $K$, making training numerically stable while retaining expressive shape control via the coefficients.

## C  EXPERIMENT DETAILS

The tables 7 and table 8 below show the hyperparameter during tuning the models.

Table 7: Hyperparameter settings for training ViT-Tiny on CIFAR-100.

| Hyperparameter | Value |
|---|---|
| Input resolution | $224^2$ |
| Epochs | 50 |
| Batch size | 256 |
| Learning rate | $1 \times 10^{-3}$ |
| Learning rate decay | Cosine |
| Optimizer | AdamW |
| Weight decay | 0.05 |
| AMP | True |

Table 8: Hyperparameter settings for instruct tuning.

| Hyperparameter | Value |
|---|---|
| Cutoff length | 1024 |
| Flash attention | auto |
| Max new tokens | 512 |
| Max samples | 1000 |
| Per-device eval batch size | 2 |
| Preprocessing workers | 16 |
| Quantization method | bnb |
| Stage | SFT |
| Temperature | 0.95 |
| Template | default |
| Top-p | 0.7 |
| Trust remote code | True |

### C.1  IMPLEMENTATION DETAILS OF BASELINE METHODS

To ensure fair comparison across all parameter-efficient fine-tuning (PEFT) methods, we adopt a unified experimental setup based on the ViT-Tiny backbone pretrained on ImageNet-1K. All methods fine-tune the classification head, and images are resized to $224 \times 224$ using a patch size of 16. Below we detail the specific hyperparameter configurations for each baseline:

- **VPT** Jia et al. (2022): We prepend 10 learnable prompt tokens of dimension 192 to the input sequence. Only the prompts and classification head are updated. The learning rate for the prompts is $5 \times 10^{-3}$, and the training schedule matches that of full fine-tuning.

- **Adapter** Houlsby et al. (2019): Adapter modules with a bottleneck dimension of 48 are inserted between each transformer block. Only adapter parameters and the classification head are updated. Learning rate is set to $1 \times 10^{-3}$.

- **LoRA** Zhang et al. (2023): LoRA modules with rank $r = 8$ are inserted into the query and value projections of each attention layer. Alpha is set to 16, and dropout is disabled. Only LoRA parameters and the classification head are updated.

- **QLoRA** Dettmers et al. (2023): The backbone is quantized to 4-bit precision using NF4 format. LoRA is applied with the same configuration as above. We use gradient checkpointing and double quantization as described in the original paper.

- **DoRA** Liu et al. (2024): All linear weights are decomposed into magnitude and direction, with both components trainable. Initialization follows the frozen pre-trained weights. The learning rate is $5 \times 10^{-4}$, consistent with the original DoRA setup.

# D    OTHER EXPERIMENT RESULTS

## D.1    OTHER ABLATION STUDIES

**Rank Setting.** We evaluate the performance of NoRA under varying amounts of tunable parameters by adjusting the rank $r$ in the low-rank updates of the rational function coefficients. Specifically, we use the ViT-Tiny model pretrained on ImageNet-1K and fix the rational function structure like GR-KAN as $(m = 5, n = 4)$. To investigate the trade-off between expressivity and parameter efficiency, we experiment with $r \in \{1, 2, 3, 4\}$ while keeping other training configurations unchanged. As shown in Figure 3, increasing the rank leads to improved performance up to $r = 3$, beyond which gains saturate. The setting $r = 3$ achieves the best accuracy on CIFAR-100, suggesting that it strikes a good balance between capacity and efficiency. Notably, when calculating the number of parameter in this task we ignore the classification head.

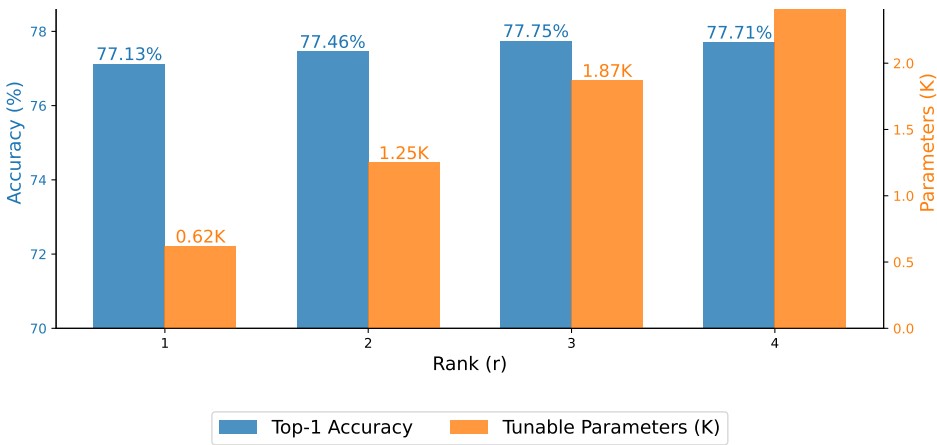

Figure 3: Accuracy-Parameter Trade-off in NoRA on CIFAR-100 with Varied Low-Rank Updates on ImageNet-1K Pretrained ViT-Tiny.

## D.2 ADAPTABILITY OF ACTIVATION FUNCTIONS WITH DIFFERENT PEFT METHODS

Before and after fine-tuning the model, if the activation distribution of the same batch of samples significantly changes across layers, it indicates that the method provides the activation function with greater *plasticity*. Conversely, if the activation distribution remains relatively stable, the adaptability is weaker. We typically use distribution distance as the core metric, normalizing it to the interval $[0, 1]$, and define the **Adaptability Score** in this context. As shown in Figure 4, mainstream methods have not yet effectively explored the adaptability of activation functions.

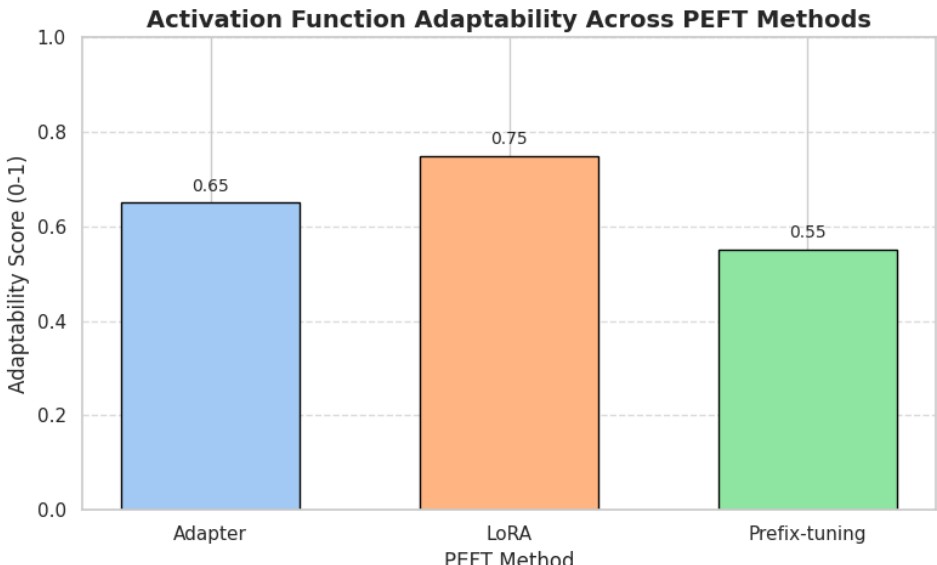

Figure 4: Comparative analysis of Different PEFT Methods in terms of the adaptability of activation functions

## D.3 ANALYSIS OF CONVERGENCE RATE

As illustrated in Figure 5, we plot the training curves of NoRA and full fine-tuning across training epochs. NoRA converges significantly faster, stabilizing around epoch 20, whereas full fine-tuning requires nearly 45 epochs to reach convergence. This demonstrates NoRA's capacity to leverage pre-trained knowledge more efficiently, leading to faster and more stable adaptation. Likewise, although our method does not outperform full fine-tuning on the training set, it achieves better generalization on the test set. This indicates that our approach imposes a beneficial inductive bias, likely mitigating overfitting and preserving useful priors from the pretrained model.

## D.4 RESULT VISUALIZATION

**t-SNE Visualization.** To further understand the representational effect of NoRA, we perform a t-SNE visualization on the learned feature embeddings for a selected subset of 10 diverse CIFAR-100 classes. As shown in Figure 6, the embeddings obtained through full fine-tuning exhibit well-separated clusters, indicating task-specific adaptation with high discriminative power. Interestingly, NoRA achieves a similarly clear cluster structure despite tuning only 0.02% of parameters, suggesting that it successfully reshapes the learned representation space within a low-dimensional subspace.

**Grad-CAM Visualization.** To intuitively demonstrate how adjusting nonlinear activations influences transferability, we visualize the final block of the ViT-Tiny model before and after fine-tuning using Grad-CAM on selected samples from CIFAR-100. As shown in Figure 7, the fine-tuned model exhibits stronger focus on the main objects within the images, indicating enhanced feature localization. This suggests that fine-tuning the activation functions effectively improves the model's performance on downstream tasks by enabling better transfer and reuse of relevant features.

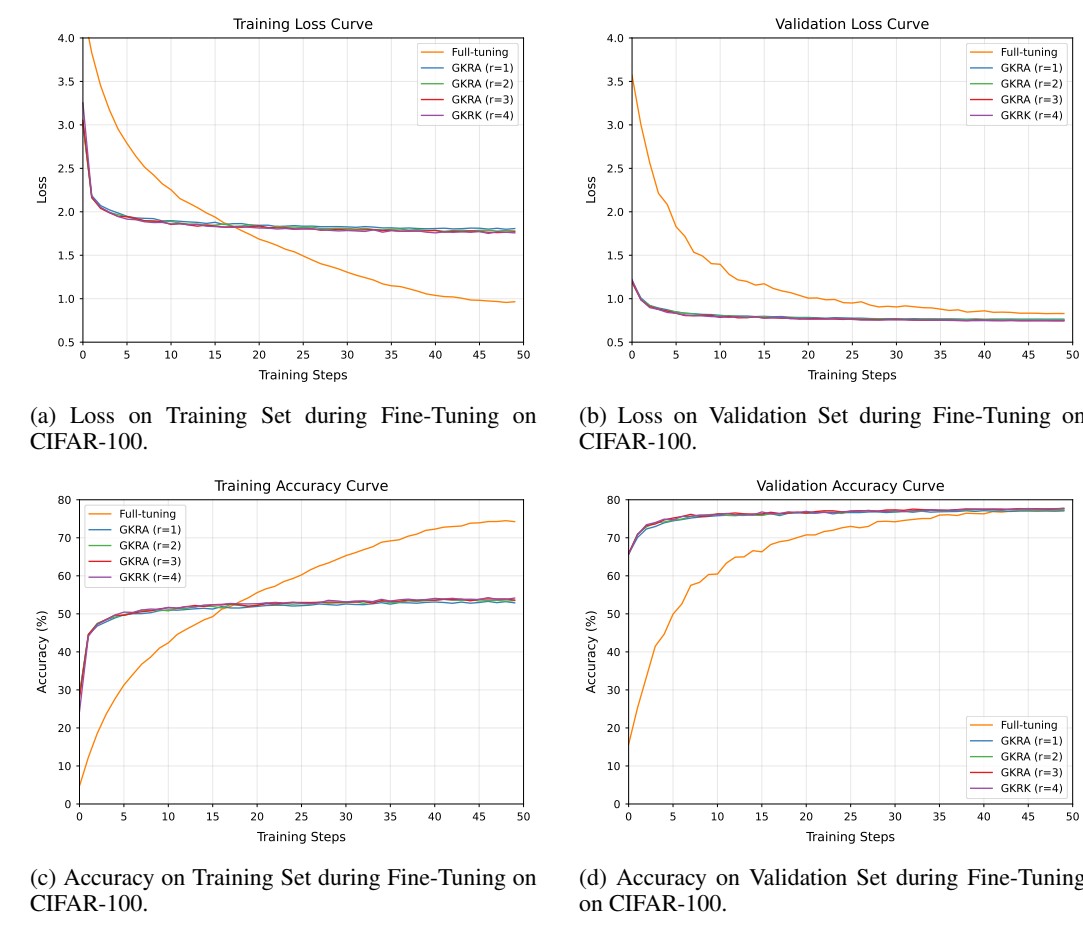

(a) Loss on Training Set during Fine-Tuning on CIFAR-100.

(b) Loss on Validation Set during Fine-Tuning on CIFAR-100.

(c) Accuracy on Training Set during Fine-Tuning on CIFAR-100.

(d) Accuracy on Validation Set during Fine-Tuning on CIFAR-100.

Figure 5: Comparison of Convergence Efficiency and Accuracy between NoRA and Full Fine-Tuning across four experimental settings.

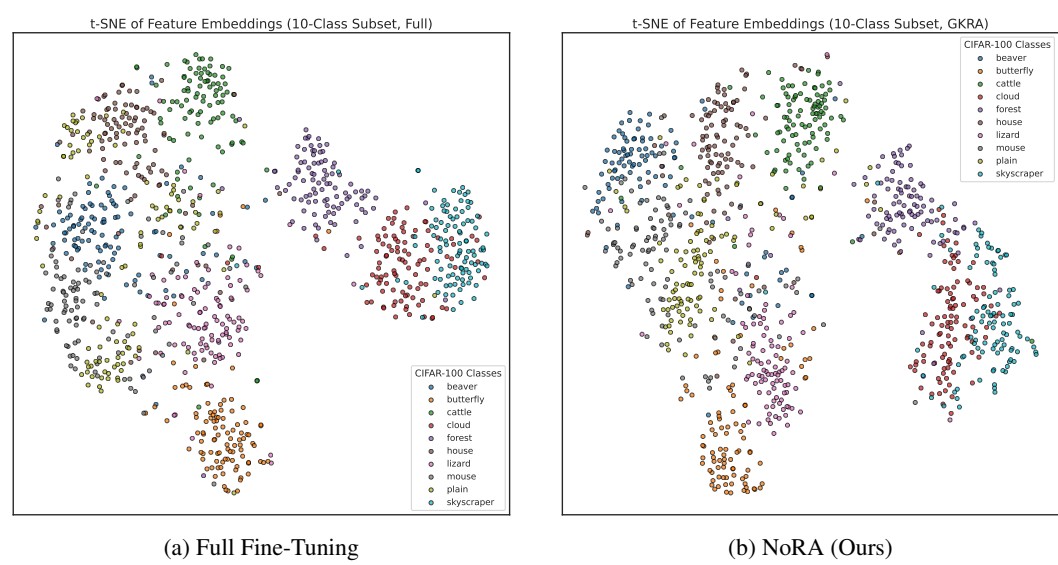

(a) Full Fine-Tuning

(b) NoRA (Ours)

Figure 6: t-SNE visualization of feature embeddings on a 10-class subset of CIFAR-100. (a) Full fine-tuning produces well-separated clusters. (b) NoRA achieves comparably structured representations with 0.02% parameter updates, illustrating its capacity to retain discriminative geometry while preserving pretrained inductive priors.

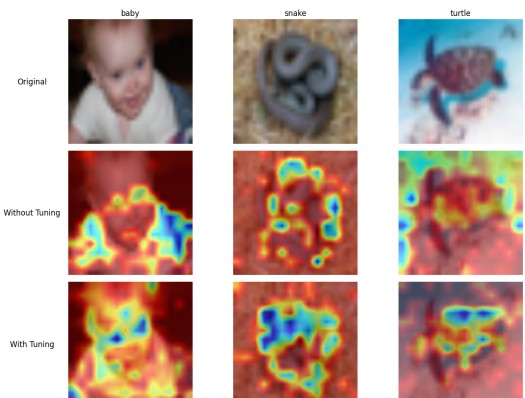

Figure 7: Grad-CAM comparison chart before and after NoRA fine-tuning

## E  CODE AVALIABLE

Our code is available in `https://anonymous.4open.science/r/NoRA_1`.

## F  THE USE OF LARGE LANGUAGE MODELS (LLMS)

Parts of this manuscript were linguistically polished with the assistance of a large language model (LLM), specifically ChatGPT (GPT-5). The model was only used for improving grammar, phrasing, and clarity. All research ideas, experimental designs, data collection, analyses, and conclusions are solely the work of the authors.

