# OpenReview forum: "Don't Forget the Nonlinearity: Unlocking Activation Functions in Efficient Fine-Tuning"
_ICLR.cc/2026/Conference — ICLR 2026 Conference Withdrawn Submission_

### Official Review · Reviewer_gphG · 2025-10-27

**Soundness:** 3
**Presentation:** 3
**Contribution:** 2
**Rating:** 4
**Confidence:** 4

**Summary:**

This paper proposes NoRA (Nonlinear Rational Adapter), a new PEFT framework that focuses on adapting activation functions rather than weight matrices.
Unlike existing methods such as LoRA which update only linear weights, NoRA introduces learnable rational functions as adaptive nonlinearities and applies structured low-rank perturbations to both numerator and denominator coefficients.
The paper demonstrates that activation-level adaptation can achieve or even surpass full fine-tuning with less than 0.5% of trainable parameters.
Extensive experiments on ViT-Tiny (CIFAR-10/100) and LLaMA3-8B instruction tuning show that NoRA and its hybrid variant NoRA++ outperform weight-based PEFT baselines under similar parameter budgets.

**Strengths:**

* Shifts PEFT from weight-centric adaptation to activation-centric adaptation, introducing a fresh and underexplored research direction.
* The paper provides detailed mathematical analysis on why activation tuning improves expressivity and stability

**Weaknesses:**

* Although the focus on activations is interesting, the core mechanism (low-rank perturbation of function coefficients) conceptually resembles LoRA applied to a new component; the leap may be seen as moderate rather than groundbreaking
* Most results are on ViT-Tiny; larger backbones (e.g., ViT-B, ViT-L) would strengthen the generality claims.
* Since rational functions involve division, potential numerical instability or exploding gradients could be further analyzed or visualized.
* It would be valuable to show comparisons to other recent sota LoRA-based approaches.

**Questions:**

* How sensitive is NoRA to the degree of the rational function (m, n)? Could over-parameterized rationals cause instability or overfitting?
* For large-scale LLMs, how does NoRA affect inference latency or memory footprint when scaled to billions of parameters?
* How does the performance change if the denominator coefficients are fixed and only the numerator is updated?

---

### Official Review · Reviewer_p9SJ · 2025-10-31

**Soundness:** 2
**Presentation:** 4
**Contribution:** 3
**Rating:** 4
**Confidence:** 3

**Summary:**

Recently, KAN networks and other model architectures have been built with learnable activation functions, as compared to the more standard choice of learning just the linear operators. In this same vein, this work now proposes a method to do the same but for fine-tuning. Specifically, rational functions are chosen to be the learned activation functions. The work then goes on to show that NoRA (the new method) can do what LoRA can do with significantly less parameters, as well as performing in tandem with LoRA to have a greater performance than with full fine-tuning. Most notably, compared to other techniques that allow for learnable activations at fine-tuning, this algorithm has superior performance.

**Strengths:**

- Compared to other methods, the experimental evidence shows that NoRA performs better with fewer parameters as compared to LoRA, and can be used to amend other fine-tuning techniques to perform better than fine-tuning. Such improvements on PEFT with fewer parameters is useful
- When compared to other existing methods that allow for the model activations to be learned, NoRA performs significantly better
- The appendix contains appealing theorems and propositions about the utility of allowing the activations to be rational functions. These theorems also cover a diverse range of concepts that could be applicable when investigating rational activation functions further
- The method very clearly addresses some clear issues of learning rational functions (namely the sensitivity around zero)

**Weaknesses:**

- The experiments for NoRA and related methods have a few missing elements (see questions). These mostly center around choices of the number of groups for experiments such as those in Table 1 as well as any experiments where NoRA is given sufficiently many groups to have comparable parameter count to LoRA
- The statements/propositions/theorems within the appendix either go without proof/citation, or they have a proof but it is loosely integrated into the text rather than being clearly written as a distinct proof. For example, there is a reference to how rational functions are able to achieve geometric convergence rather than algebraic convergence for polynomials. The is likely true, but a citation is needed

**Questions:**

- It isn't clear how many groups are used in the experiments in Table 1
- In the experiments, especially Table 1, what is being held constant between the different methods? Are they all being trained to convergence, or is the compute the same, or is the number of epochs the same? "Training budgets" are mentioned, but the precise meaning of that would be appreciated

- Are the results averaged across multiple runs with different random initializations? It's currently hard to tell if these results in Table 2 are truly outperforming LoRA or if they are due to randomness (including standard deviations across runs would be very helpful)

- Is it possible to instead hold the number of parameters in LoRA and NoRA equal, where a greater number of groups would give NoRA greater expressivity/parameter-count?
- Section 4.4.2, why are Rational FT and Zero-init FT different? Shouldn't these learn the same thing, or are you saying there is a distinction because of things like weight decay?

---

### Official Review · Reviewer_5F7y · 2025-10-31

**Soundness:** 2
**Presentation:** 2
**Contribution:** 2
**Rating:** 2
**Confidence:** 5

**Summary:**

This paper presents NoRA (Nonlinear Rational Adapter) — a PEFT method that departs from the usual weight-space updates (as in LoRA, DoRA, etc.) by targeting activation functions. It replaces fixed activations (e.g., GELU) with learnable rational functions and applies structured low-rank perturbations to both numerator and denominator coefficients in a group-wise manner. The proposed NoRA++ combines this activation-space tuning with LoRA. The proposed method is tested using ViT-Tiny on CIFAR10/100, and LLaMA3 8B on instruction tuning with slightly improvement against baseline such as LORA.

**Strengths:**

+ It is interesting to explore explicit activation-space PEFT methods, moving the locus of adaptation from weights to nonlinear functions — a clean conceptual contribution.

+ Theoretical appendices (Sec. A–B) justify why activation tuning matters, offering Jacobian, NTK, and Lipschitz-based insights.

**Weaknesses:**

- While the activation-centric perspective is fresh for PEFT, the method is structurally simple: substituting activations with rational functions and LoRA-style perturbations. The rational activation concept (e.g., KAN, Group-KAN, RationalNet) is not new, and NoRA mostly reuses these ideas with LoRA-style grouping.

- The proposed method is not sufficiently evaluated. It's not clear why the ViT-Tiny on CIFAR10/100 is chosen as vision experiments under the PEFT context.  On LLaMA3-8B, MMLU improvements are small (≤ 0.8%) and inconsistent (some negative deltas in Humanities and “Other” categories). No resource or convergence analysis is shown for these large-scale runs — leaving unclear if gains justify the added complexity. The evaluations do not provide GPU memory footprint, training wall-time in fine-tuning large models.

- The proposed theory (Append. A) is not empirically validated. There are no metrics on Lipschitz control, NTK diversity, or curvature modulation to substantiate claims. This disconnect weakens the theoretical claims’ impact.

**Questions:**

Please consider to address questions in the weaknesses.

---

### Note · Authors · 2026-01-06

I have read and agree with the venue's withdrawal policy on behalf of myself and my co-authors.